# Intraparenchymal and Subarachnoid Hemorrhage in Stereotactic Electroencephalography Caused by Indirect Adjacent Arterial Injury: Illustrative Case

**DOI:** 10.3390/brainsci13030440

**Published:** 2023-03-04

**Authors:** Toshiyuki Kawashima, Takehiro Uda, Saya Koh, Vich Yindeedej, Noboru Ishino, Tsutomu Ichinose, Hironori Arima, Satoru Sakuma, Takeo Goto

**Affiliations:** 1Department of Neurosurgery, Osaka Metropolitan University Graduate School of Medicine, Osaka 545-8585, Japan; 2Division of Neurosurgery, Department of Surgery, Thammasat University Hospital, Faculty of Medicine, Thammasat University, Pathumthani 12120, Thailand; 3Department of Pediatrics, Osaka Metropolitan University Graduate School of Medicine, Osaka 545-8585, Japan

**Keywords:** stereotactic electroencephalography, hemorrhagic complication, intraparenchymal hemorrhage, arachnoid trabeculae, pseudoaneurysm

## Abstract

The complication rate of stereotactic electroencephalography (SEEG) is generally low, but various types of postoperative hemorrhage have been reported. We presented an unusual hemorrhagic complication after SEEG placement. A 20-year-old man presented with suspected frontal lobe epilepsy. We implanted 11 SEEG electrodes in the bilateral frontal lobes and the left insula. Computed tomography after implantation showed intraparenchymal hemorrhage in the left temporal lobe and insula and subarachnoid hemorrhage in the left Sylvian cistern. Later, the point of vessel injury was revealed from the identification of a pseudoaneurysm, but this location was not along the planned or actual electrode trajectory. The cause of hemorrhage was suggested to be indirect injury from stretching of the arachnoid trabeculae by the puncture needle.

## 1. Introduction

Stereotactic electroencephalography (SEEG) is an epoch-making method of depth electrode insertion for intracranial EEG evaluation in epilepsy surgery and has recently spread worldwide as a key method of intracranial EEG evaluation [1,2,3,4,5]. Depth electrodes can evaluate deeper parts of the brain such as insula, operculum, and cingulate gyrus compared with subdural electrodes which are implanted to cover some part of the brain surface. The complication rate from SEEG has been thought to be low, but hemorrhagic complications are reported in 19.1% of patients and symptomatic hemorrhaging in 2.2% [6]. Another study summarized complication rates by three hemorrhagic types: epidural hemorrhage (0.3%), subdural hemorrhage (0.4%), and intraparenchymal hemorrhage (0.7%) [7]. Therefore, surgeons should not ignore the possibility of hemorrhagic complications. Here, we report an unusual case of intraparenchymal hemorrhage in SEEG. Pre- and postoperative evaluations supported the hemorrhage being caused by an indirect injury to the adjacent artery after the arachnoid trabeculae was stretched by the puncture needle.

## 2. Clinical Presentation

A 20-year-old, left-handed man presented with focal impaired awareness seizure, sometimes progressing to secondary generalized seizures with his neck rotated to the right side. His seizure started at the age of eight and had not been controlled despite taking multi antiseizure medicines (valproic acid: 700 mg, levetiracetam: 3000 mg, perampanel: 8 mg). Interictal scalp EEG showed frequent spikes and waves in bilateral frontal and temporal areas (left > right) (Figure 1A). Ictal EEG suggested a seizure origin in the left frontal lobe. In a Wada test, the patient showed no signs of aphasia when propofol was injected into each internal carotid artery, suggesting that language processing was performed in both hemispheres. Magnetic resonance imaging (MRI) showed no obvious abnormalities (Figure 1B) and ^18^F-fluorodeoxyglucose-positron emission tomography showed slight low accumulations in some parts of the left frontal lobe, particularly the basal part (Figure 1C). Computed tomography (CT) angiography showed no arterial and venous anomaly. Higher brain function score on the Wechsler adult intelligence scale IV (WAIS-IV) was 72 for full scale intelligence quotient (FIQ), 67 for verbal comprehension index (VCI), 82 for perceptual reasoning index (PRI), 91 for working memory index (WMI), and 75 for processing speed index (PSI).

We hypothesized that the seizures were originating from the antero-basal part of the left frontal lobe and propagating dorsally and contralaterally. Intracranial EEG evaluation with SEEG was scheduled. We planned to implant 11 electrodes in the bilateral frontal lobes and the left insula. Our plan was designed to avoid crossing any arteries or veins using source images from CT arteriography and venography (Figure 2 and Table 1). The patient consented to the procedure. 

Using robotic arm guidance with Stealth Autoguide (Medtronic, Minneapolis, MN, USA), we placed all electrodes under general anesthesia. Our surgical procedure of SEEG implantation was described in detail elsewhere [8]. In brief, after adjusting each trajectory of the insertion by Stealth Autoguide, a small skin incision was made followed by a skull perforation with 2.4 mm drill (Medtronic). Dura under the skull was intentionally perforated by the drill. The depth of the puncture was automatically calculated by Stealth Autoguide. Then, a puncture of the brain was performed using a needle 2.1 mm in diameter. After removing the needle, we inserted a depth electrode 1.5 mm in diameter (Unique Medical, Tokyo, Japan). We checked the depth of the brain puncture and electrode insertion with intraoperative fluoroscopy. After SEEG implantation, the patient demonstrated moderate disturbance of consciousness and right hemiparesis, but no aphasia. CT showed intraparenchymal hemorrhage in the temporal lobe and insula and subarachnoid hemorrhage in the Sylvian cistern on the left side (Figure 3A–C). This hemorrhage was located around the electrode inserted to the insula from the precentral gyrus (Electrode #9) (Figure 4). We immediately performed hematoma evacuation with fronto-temporal craniotomy and electrode removal (Figure 3D–F). Intraoperatively, any bleeding points were found in and around the hematoma cavity. 

After the surgery, the level of consciousness and hemiparesis recovered well, but CT angiography 14 days after SEEG demonstrated an aneurysmal formation at the distal middle cerebral artery (MCA), which was considered to represent the point of arterial injury. A superimposed image of the 14-day postoperative CT angiography and preoperative MRI showed that an aneurysm was located at the opercular segment of MCA (Figure 5A,B). A superimposed image of the preoperative CT angiography and postoperative CT showed that the actual trajectory of the electrode was apart from the MCA (Figure 5C). A superimposed image of the 14-day postoperative CT angiography and postoperative CT showed a postoperative mild deviation of the running course of MCA, but the actual trajectory was apart from the location of the aneurysm (Figure 5D). 

These findings demonstrated that the aneurysm was located along neither planned nor actual trajectories of electrodes, suggesting that the cause of hemorrhage might have been an indirect injury to the MCA. We performed emergency surgery for this aneurysm. After opening the sylvian fissure, the M2 portion of MCA was found at the surface of the insula. Following MCA distally, the aneurysm was found at the M3 and M4 portion of the MCA (Figure 6A). The proximal flow was temporarily arrested by a clip. A good collateral flow to the distal part was confirmed with indocyanine green injection (Figure 6B). Then, we performed internal trapping of the aneurysm and removed it (Figure 6C,D).

Pathological findings showed that the aneurysm was a pseudoaneurysm. Postoperatively, the patient returned to daily life after rehabilitation without any neurological deterioration. He has not experienced any seizures since then. WAIS-IV at 1 year after surgeries was 82 for FIQ, 71 for VCI, 114 for PRI, 69 for WMI, and 69 for PSI.

## 3. Discussion

Postoperative hemorrhage is the most concerning complication of SEEG. Although complete removal of risk is impossible, surgeons should make the maximum effort to reduce the risk of hemorrhage. Preoperatively, all trajectories must be planned to avoid crossing any visible arteries or veins on CT angiography and/or MR angiography. In particular, arteries and veins running in the subarachnoid space show a higher risk of injury than small medullary vessels running in the brain parenchyma, because vessels in the subarachnoid space are fixed to the pia mater by the arachnoid trabeculae. The usual cause of vessel injury is direct damage by the puncture needle. However, another possible but unusual cause, as in this case, is indirect damage caused by excessive stretching of the arachnoid trabeculae. Along this line, one factor increasing the risk of hemorrhage is likely the number of pial penetrations. We speculated that the cause of hemorrhage in this patient was indirect arterial injury, supported by the fact that the location of arterial injury in the subarachnoid space was apart from the planned or actual trajectory of the adjacent electrode. 

Generally, investigation of the insula by depth electrodes is achieved using either an orthogonal or oblique trajectory [9,10,11,12]. With the orthogonal trajectories, electrodes are inserted through the operculum part, which enables evaluation of the operculum [9]. However, the pia mater is penetrated three times. In this patient, we selected orthogonal trajectory, penetrating the pia mater at the precentral gyrus, frontal operculum, and insular cortex. In the Sylvian cistern, thick vessels are fixed to the pia mater by tough arachnoid trabeculae, which should be considered on the selection of the trajectory. On the other hand, with the oblique trajectories, electrodes are inserted from the superior frontal gyrus or superior parietal lobule to the insula [11,12]. The number of electrode contacts placed in the insular cortex is greater than with orthogonal trajectories and the pia mater is only penetrated once in this trajectory. Theoretically, oblique trajectories might be safer than orthogonal trajectories, but low complication rates have been reported with both trajectories [9,11,12] and no comparative studies have clarified the risks of hemorrhage. Although trajectories are determined depending on the specific preoperative information in each patient and the preferences of each institution, we now prefer oblique trajectory to the insula because of the theoretically safer profile after encountering the complication in this case.

To penetrate the pia mater without excessive stretching of the arachnoid trabeculae, the use of thin puncture needles would be safer. Thinner puncture needles and electrodes are not currently available in Japan; the puncture needle used for this patient was 2.1 mm in diameter and the electrodes were 1.5 mm in diameter. Once thinner electrodes and puncture needles are introduced, the risk of excessive stretching of the arachnoid trabeculae will presumably decrease.

Fortunately, this patient recovered well from the neurological deficit and returned to his daily life as a university student. Higher brain function score on WAIS-IV did not decline postoperatively. This is possibly because the patient’s language function was processed bilaterally which was proved with a WADA test. Furthermore, the patient has not experienced any seizures since surgery. Obviously, that was an unexpected favorable result, which probably occurred because the epileptogenic zone or network was totally involved in the damaged brain by the hemorrhage. By February 2023, eighteen months had passed since the surgery, but we are planning to continue following up this patient carefully.

## 4. Conclusions

We reported an unusual complication of intraparenchymal and subarachnoid hemorrhage following SEEG. Pre- and postoperative evaluations suggested that this hemorrhage was caused by excessive stretching of the arachnoid trabeculae resulting in indirect injury to an adjacent artery. This report offers a caution to epilepsy surgeons regarding the possibility of indirect vessel injury in the subarachnoid space and the potential risk of thick puncture needles.

## Figures and Tables

**Figure 1 brainsci-13-00440-f001:**
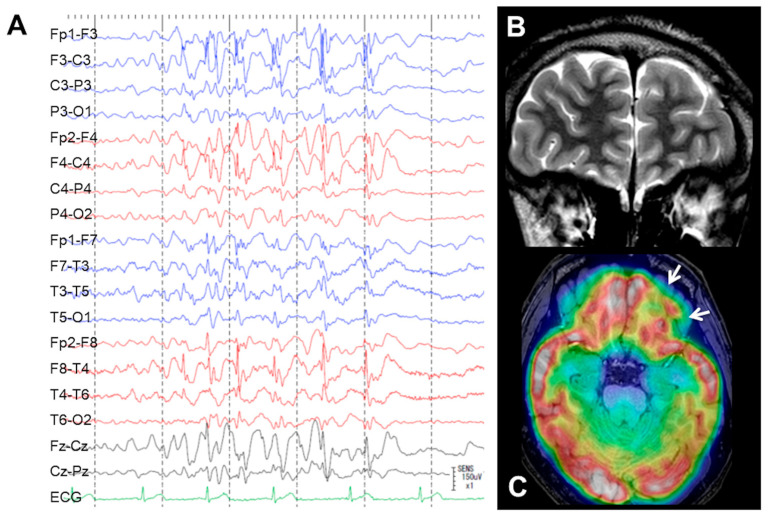
(**A**) Interictal scalp electroencephalography shows frequent spikes and waves in bilateral frontal and temporal areas (left > right). (**B**) Magnetic resonance imaging shows no obvious abnormalities. (**C**) Slight low accumulation is seen on ^18^F-fluorodeoxyglucose-positron emission tomography at the basal part of the left frontal lobe (white arrow).

**Figure 2 brainsci-13-00440-f002:**
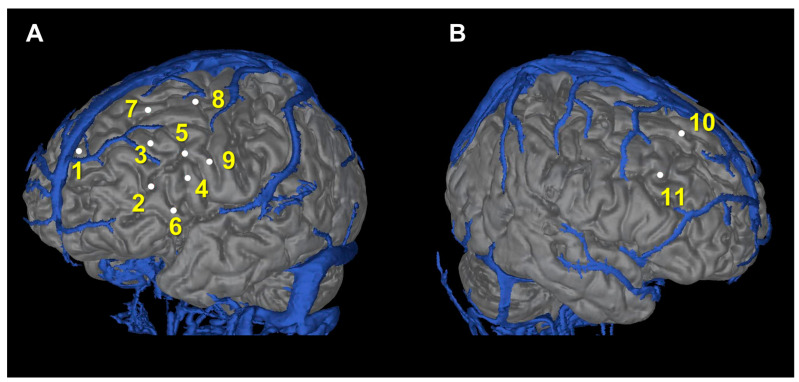
Planning for electrode implantation. A total of 11 electrodes were planned for bilateral frontal lobes and the left insula. Yellow numbers correspond to the electrode numbers in Table 1. Right side (**A**); Left side (**B**).

**Figure 3 brainsci-13-00440-f003:**
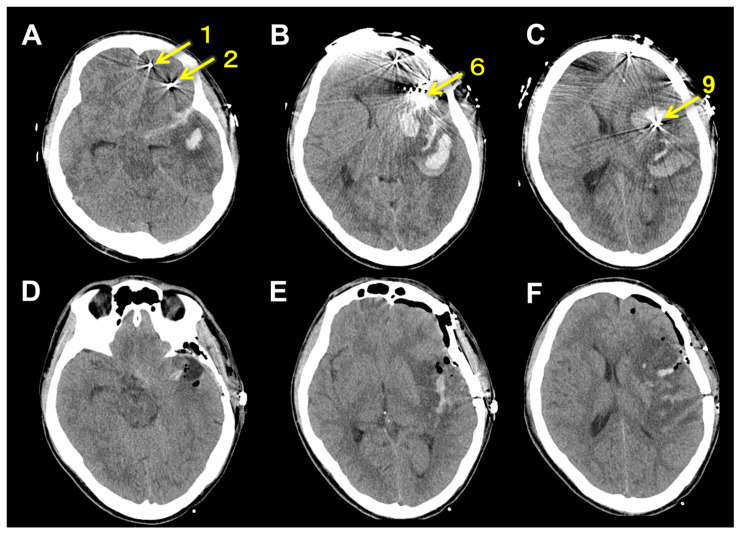
(**A**–**C**): Computed tomography shows intraparenchymal hemorrhage in the temporal lobe and insula and subarachnoid hemorrhage in the Sylvian cistern on the left side. Yellow numbers correspond to the electrode numbers in Table 1. (**D**–**F**): Computed tomography just after hematoma evacuation and electrode removal. Almost all the intraparenchymal hemorrhage has been removed.

**Figure 4 brainsci-13-00440-f004:**
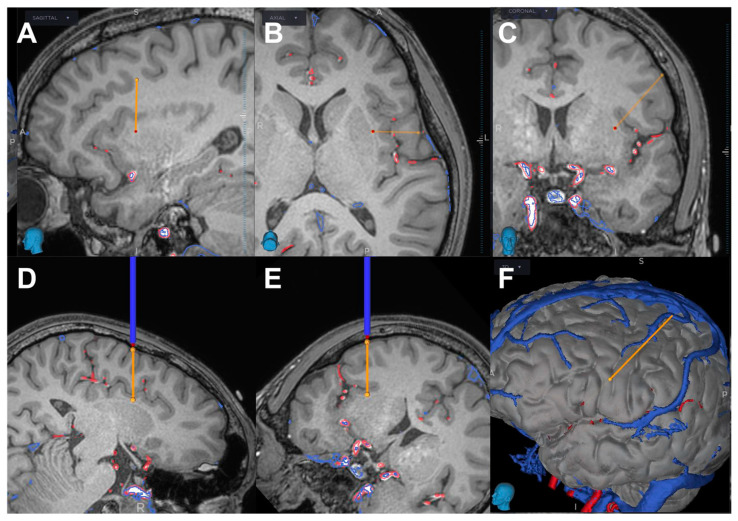
Implantation plan of Electrode #9 (orange and blue line). The trajectory is planned from the left precentral gyrus to the insula, not crossing any arteries or veins. Sagittal (**A**), axial (**B**), coronal (**C**), trajectory views (**D**,**E**), and 3-dimensional (**F**) appearances.

**Figure 5 brainsci-13-00440-f005:**
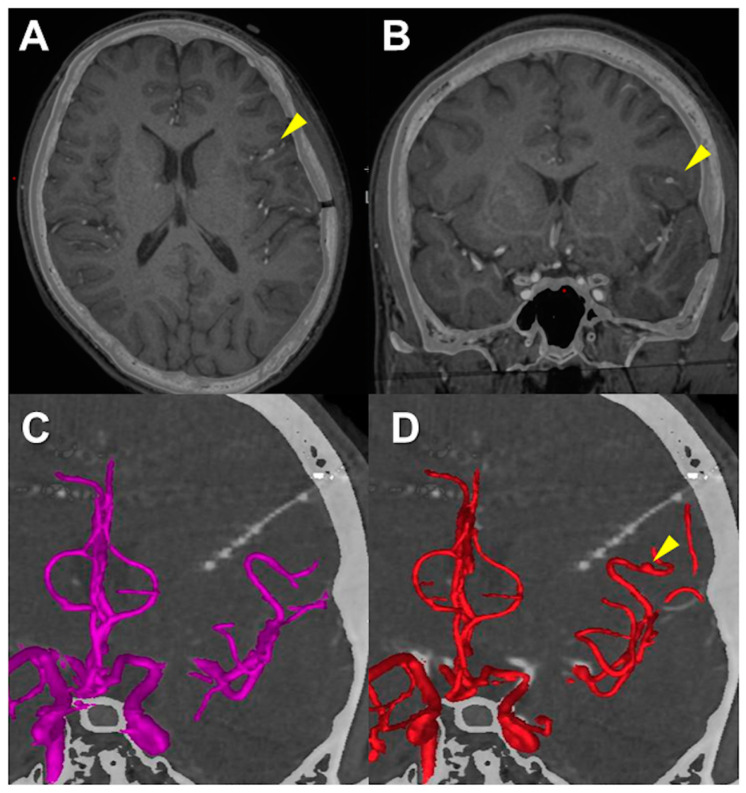
(**A**,**B**) Superimposed image of 14-day postoperative computed tomography angiography and preoperative magnetic resonance image. (Axial (**A**) and coronal (**B**) sections) Yellow arrowheads show the location of the aneurysm. (**C**) Superimposed image of preoperative computed tomography angiography and postoperative computed tomography. The actual trajectory of Electrode #9 is apart from the middle cerebral artery. (**D**) Superimposed image of 14-day postoperative computed tomography angiography and postoperative computed tomography. The actual trajectory of Electrode #9 is apart from the location of the aneurysm (yellow arrowhead) which is suggested as the point of arterial injury.

**Figure 6 brainsci-13-00440-f006:**
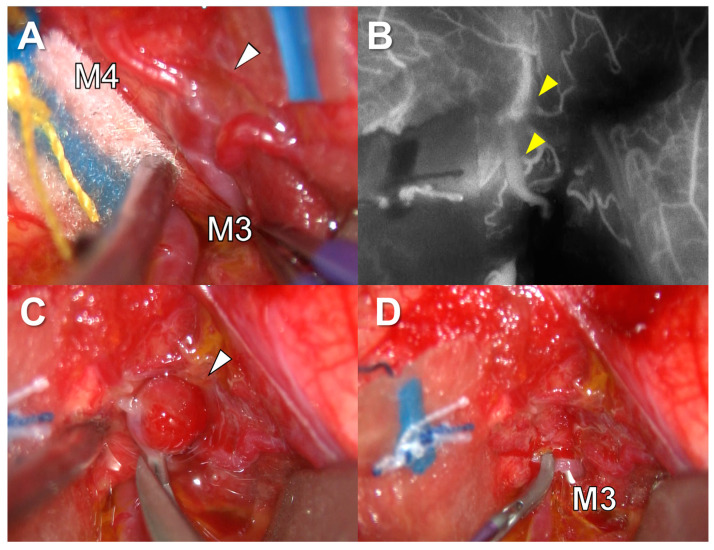
(**A**) The aneurysm at M3 and M4 portion of the middle cerebral artery. White triangle indicates the aneurysm. (**B**) Confirmation of the good collateral flow to the distal part (yellow triangle) with indocyanine green injection. (**C**) Internal trapping of the aneurysm (white triangle). (**D**) Final surgical view after the removal of the aneurysm.

**Table 1 brainsci-13-00440-t001:** Location of the electrodes.

Electrode #	Side	Entry	Target
1	Left	SFG	anterior Orb
2	Left	IFG	posterior Orb
3	Left	MFG	anterior Cing
4	Left	IFG	anterior Cing
5	Left	FEF	middle Cing
6	Left	IFG	middle Cing
7	Left	anterior SFG	SMA
8	Left	posterior SFG	SMA
9	Left	PreCG	insula
10	Right	SFG	SMA
11	Right	MFG	middle Cing

Cing: cingulate gyrus; FEF: frontal eye field; IFG: inferior frontal gyrus; MFG: middle frontal gyrus; Orb: orbital gyrus; PreCG: precentral gyrus; SFG: superior frontal gyrus; SMA: supplementary motor area.

## Data Availability

Not applicable.

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
