# Peer review of "Intraparenchymal and Subarachnoid Hemorrhage in Stereotactic Electroencephalography Caused by Indirect Adjacent Arterial Injury: Illustrative Case"

_brainsci, 2023, doi:10.3390/brainsci13030440_

Round 1

Reviewer 1 Report

- This is a reasonably well-written manuscript detailing the potential for complications from intracranial stereoelectroencephalography (sEEG).

- Overall, the clinical case presentation and images are excellent. There is an assumption made that the pseudoaneurysm was formed by 'stretching of arachnoid trabeculae.' I would mention this comment as a possible scenario in the discussion, as the authors have already discussed, but would recommend they remove this short phrase from the rest of the title (i.e. keep the title as follows -- 'Intraparenchymal and subarachnoid hemorrhage in stereotactic electroencephalography caused by indirect adjacent arterial injury: illustrative case.'

- Line 124 - Recommend mentioning the utility of MRA for additional pre-operative planning to mitigate against vascular injury (in addition to CTA), where available.

- I would revise line 148-149 to reflect that:  Both oblique vs orthogonal electrodes likely feature a similar/comparable risk for hemorrhage, while optimizing implantation of the insula.

- The authors should include mention of the specific Robotic company and Electrode/equipment company used. Was it ROSA vs Renishaw vs some other company? DIXI vs Adtech vs PMT vs other?

Reviewer 2 Report

Thank you for thise illustrative case report. However, I suggest some changes/improvements:

- the text would be benefit from an english editing

Introduction

-         The statement “The complication  rate from SEEG appears low, but hemorrhagic complications are reported in 19.1% of patients” sems paradoxical as a hemorrhagic rate of 19,1% is not low.

Clinical Presentation

-         There is no information on how long the epilepsy has existed and what medications the patient is receiving.

-         The representation of the entry points of the electrodes in figure 2 seems somewhat unusual from a stereotactic point of view: electrodes 2, 4, 5, and 6 appear to enter sulci and not gyri of the brain. This significantly increases the risk of hemorrhage and raises the question of whether this is reason for the hemorrhage. The authors should comment on this

-         Which robotic arm/system was used?

-         Please label electrodes in Fig. 3: in Fig. 3A readers would assume that the causative electrode is located anterior to the insula, while there is another electrode in Fig. 3C within the insula. However, Fig. 4 and 5 suggest that maybe the electrode from Fig. 3A is located nearest tot he pseudoaneruysm. A final assessment is difficult without a 3D representation, but maybe you can present these electrodes also in Fig. 4/5.

-         The patient had no seizures since bleeding? Is this correct?

Round 2

Reviewer 2 Report

Dear authors,

thank you for your corrected version of the manuscript.

I agree with not modifying Fig. 2, although I still think that the demonstrated entry point could be critically reviewed by stereotactic neurosurgeons.

However, to better understand the positional reference of the electrode, I expressly request to add inline views/probe views of the electrode in Fig. 4.

Kind regards

Author Response

Thank you for your consideration for our manuscript.

As you suggested, I add inline views of the electrode #9 in Fig.4.

Probe view was not so understandable.